# Gene expression and metabolic changes of *Momordica charantia* L. seedlings in response to low temperature stress

Yu Niu[◉], Ziji Liu[◉], Huang He, Xu Han, Zhiqiang Qi*, Yan Yang*

Tropical Crops Genetic Resources Research Institute, CATAS, Danzhou Hainan, China

◉ These authors contributed equally to this work.
* yziqi@126.com (YY); zhiqiangqi@126.com (ZQ)

## Abstract

Low temperature is one of the abiotic factors limiting germination, growth and distribution of the plant in current plant-products industry, especially for the tropical vegetables in non-tropical area or other fields under cold temperature. Screening the plant with ability against cold temperature captured worldwide attention and exerted great importance. In our previous work, the anti-cold specie of *Momordica Charantia* L. seedlings was screened out. Yet, the molecular and physiological mechanisms underlying this adaptive process still remain unknown. This study was aimed to investigate adaption mechanism of anti-cold species of *Momordica Charantia* L. seedlings in genetical and metabolomics levels. Two species, cold-susceptible group (Y17) and cold-resistant group (Y54), were evaluated containing the indexes of malondialdehyde (MDA), hydrogen peroxide ($H_2O_2$), proline content, activities of antioxidant enzymes, metabolites changes and genes differentiation in plant tissues after cold treatment. It found that low temperature stress resulted in increased accumulation of MDA, $H_2O_2$ and proline content in two species, but less expressions in cold-resistant species Y54. As compared to Y17, cold-resistant species Y54 presented significantly enhanced antioxidant enzyme activities of POD (peroxidase), CAT (cataalase) and SOD (superoxide dismutase). Meanwhile, higher expressed genes encoded antioxidant enzymes and transcription factors when exposure to the low temperature were found in cold-resistant species Y54, and core genes were explored by Q-PCR validation, including McSOD1, McPDC1 and McCHS1. Moreover, plant metabolites containing amino acid, sugar, fatty acid and organic acid in Y54 were higher than Y17, indicating their important roles in cold acclimation. Meanwhile, initial metabolites, including amimo acids, polypeptides, sugars, organic acids and nucleobases, were apparently increased in cold resistant species Y54 than cold susceptible species Y17. Our results demonstrated that the *Momordica Charantia* L. seedlings achieved cold tolerance might be went through mobilization of antioxidant systems, adjustment of the transcription factors and accumulation of osmoregulation substance. This work presented meaning information for revealing the anti-cold mechanism of the *Momordica Charantia* L. seedlings and newsight for further screening of anti-cold species in other plant.

**Funding:** This work was financially supported by National Nonprofit Institute Research Grant of CATAS-TCGRI (1630032017027), Key Research

and Development Project of Hainan Province
(ZDYF2019103). The funders had no role in study
design, data collection and analysis, decision to
publish, or preparation of the manuscript.

**Competing interests:** The authors have declared
that no competing interests exist.

## Introduction

Low temperature is the primary abiotic factor limiting germination, growth and distribution of the plant [1] [2]. When subjected to low temperature stress, plants usually evolve various physiological and biochemical changes, and modulating the expression of gene to achieve cold acclimation, such as increasing antioxidant enzymes antioxidants, and osmotic solutes [3] [4]. Low temperature stress induces enhanced accumulation of free radical [5], destroys the dynamic equilibrium of activate oxygen metabolism [6] [7], and decreased osmotic solutes [8] in plants. Through maintaining active oxygen scavenging system, reducing the toxicity of oxygen free radical to the cell and decreasing the destruction in cell structure, cold tolerance of the plant was improved [9]. Meanwhile, low temperature could also cause the accumulation of osmoregulation substances which could increase intracellular solute concentration, lower ice point of the cell, and then reduce the injury of the low temperature to the cell [10] [11]. Exploring the species against cold temperature was of great importance, such as in the field of agriculture, *ect*.

Bitter gourd (*Momordica charantia* L.), a member of the Cucurbitaceae family, is a popular and widely grown vegetable in Southeast Asia due to its high-level edible and medicinal values. It contains plenty of minerals, vitamin C and E [12], *etc*. It was also applied to modulate blood glucose and treat diabetes in our daily life [13]. With increasing demand of bitter gourd in the Chinese market, plant regions of bitter gourd were gradually extended from south to north in order to meet the customers' requirement. However, the low temperature in spring and winter or most areas of the China seriously restricted germination and growth of bitter gourd since it is a tropical vegetable which is sensitive to chilling [7]. Thus, it had urgent need to breed some cultivars tolerance to low-temperature to enlarge cropping areas and guarantee market supplies. In our previous work, the anti-cold species of bitter gourd seedling was found [14]. However, the mechanism of their anti-cold ability was still unsolved.

Recently, transcriptome and metabolomics analysis had been extensively used in fields of crops to explore mechanism of cold resistance, such as rice, wheat and tomato [15] [16]. By using the transcriptome method, researcher found that fatty acid desaturases (FADs) play important roles in regulating fatty acid composition and maintaining membrane fluidity under temperature stress, and the expression of genes encoding FADs (ZmFAD2.1&2.2, ZmFAD7, and ZmSLD1&3) in maize were significantly up-regulated under cold stress [17]. Meanwhile, researcher, using the transcriptome analysis, found that tomato plants improved their tolerance of low temperature by a chilling acclimation process entailing comprehensive transcriptional and metabolic adjustments, including the accumulation of compatible solutes and mobilization of antioxidant systems [3]. Thus, transcription and metabolism method were used in this study to explore cold acclimation mechanism of the bitter gourd seedlings. Cold-susceptible bitter gourd seedlings (Y17) and cold-resistant ones (Y54) were cultivated and used to reveal anti-could mechanism of cold-resistant bitter gourd. These data could provide meaningful and practical information on understanding physiological mechanisms of bitter gourd underlying this adaptive response, and a theoretical and practical basis was also provided in this work for breeding low-temperature tolerant cultivars.

## Materials and methods

### Plant materials and growth conditions

In our pre-experiment, six bitter gourds could be divided into three cold tolerance grades under 5˚C for 1 day, and the maximum cold damage index was ranged from 20.31 to 84.38. Thus, the temperature of 5˚C could be used as the temperature index for identification of cold

tolerant species. Two bitter gourd cultivars (cold-susceptible Y17 and cold-resistant Y54) were screened out in our previous work [14] and used as the materials in this study. The experiment was carried out in 8th Solar Greenhouses of the melon and vegetable research laboratory of Tropical Crops Genetic Resources Research Institute (Danzhou, China). The bitter gourd seeds, with the same size, were subjected to hot-water treatment and cup seedling (8cm*8cm). After germination, bitter gourd seedlings were grown in pots filled with sterilized soil, and they were kept in a plant growth chamber under 12 h photoperiod with 60±5% relative humidity, temperature 25±1% and 150 μmol photons $m^{-2}$ $s^{-1}$ light intensity. When the seedlings grew to three leaves and one heart period, uniform bitter gourd seedlings (the number of each cultivar was 70) were placed in a growth chambers and subjected to low temperature treatment (5 ±1˚C) with light intensity of 150 μmol $m^{-2}$ $s^{-1}$ and 12 h photoperiod for 24 h. The leaves of six random samples of each bitter gourd cultivar were harvested after 0, 1, 2, 4, 8, 12 and 24 h of low temperature treatment. Leaf samples were either used as fresh or were immediately frozen in liquid nitrogen and stored at -80˚C before measurements.

## Determination of physiological indices at different time interval

About 0.2 g fresh leaf segments collected from each pot at different time intervals were used for the determination of proline, superoxide dismutase (SOD), (peroxidase) POD, catalase (CAT), hydrogen peroxide ($H_2O_2$) and malondialdehyde (MDA). The proline content was quantified by using a sulfosalicylic acid assay. CAT activity was detected according to reported method [7]. Concentration of MDA was assessed by a thiobarbituric acid colorimetric method [2]. POD activity was determined by the method described in previous work [16]. SOD activity was estimated by the method of nitroblue tetrazolium assay [18]. $H_2O_2$ content was assayed by the method described in reported work [18]. Meanwhile, superoxide anion kits (Solarbio Life Scince Co., Ltd., China) and hydroxyl radical kits (Nanjing Jiancheng Technology Co., Ltd., China) were used to detected O2- and ·OH according to the manufacturer's guidelines.

## Gene expression analysis

For RNA extraction and cDNA synthesis, total RNA was extracted from fresh leaves by using trizol reagents (Invitrogen, Carlsbad, CA, USA) according to manufacturer's protocols, and DNase I (Roche) was used to purify total RNA. After extraction, RNA samples were utilized to construct the cDNA libraries, each library being generated from an equivalent mixture of three independent RNA preparations. Constructing the libraries and subsequent sequencing was performed by Illumina HiSeq TM 2000 technology supplied by NovoGene corporation (Beijing, China).

Quantification of genes: HTSeq v0.6.1 was used to count the reads numbers mapped to each gene. And then FPKM of each gene was calculated based on the length of the gene and reads count mapped to this gene. FPKM, expected number of Fragments Per Kilobase of transcript sequence per Millions base pairs sequenced, considers the effect of sequencing depth and gene length for the reads count at the same time, and is currently the most commonly used method for estimating gene expression levels [19].

Differentiation of expressed genes: Differential expression analysis of two groups was performed by using the DESeq R package (1.18.0). DESeq provide statistical routines for determining differential expression in digital gene expression data by using a model based on the negative binomial distribution. The resulting *P*-values were adjusted using the Benjamini and Hochberg's approach for controlling the false discovery rate. Genes with an adjusted *p*-value <0.05 found by DESeq were assigned as differentially expressed. Genes were validated by Q-PCR analysis.

## Metabolite extraction and metabolite profiling analysis

For each bitter gourd variety, the leaves of six random samples at 0 h, 8 h and 24 h after the treatment were chosen to analyze their initial metabolites' changes. After freeze-drying, the leaf tissue was grinded to powders. Approximately 150 mg tissue powders were transferred into 2 ml centrifuge tubes. 1mL mixed solution contained methanol, acetonitrile and water (2:2:1, *v/v/v*) was added to the tubes. The mixtures were vortexed at low temperature (4˚C) for 30 min, twice. Subsequently, the tubes were placed at -20˚C for 1 h. The supernatant was decanted into a new glass tube and freeze-dried after centrifuging at 13,000 rpm (4˚C) for 15 min. Then, the samples were saved under -80˚C condition. Before GC-MS analysis, 100 uL of mixed solution contained acetonitrile and water (1:1, v/v) to the samples to vortex. After centrifuging at 14,000 rpm (4˚C) for 15 min, the supernatant was extracted to analysis. The samples were placed in the 4˚C autosampler during all analysis process. Agilent 1290 Infinity LC Systems was used to separate and chromatographic column HILIC was used to separate. The injection volume was 2 uL, the flow rate was 0.3 mL/min, and the column temperature was 25˚C, respectively. The mobile phase A was water consisted of 25 Mm ammonium acetate and 25 Mm ammonia. Mobile phase B was acetonitrile.

Original data was converted into the format of mzXML by ProteoWizard. Then XCMS program was used to extract peak area. Metabolites were identified by searching the database established by our laboratory. SIMCA-P 14.1 software package (Umetrics, Umea, Sweden) was conducted a principal component analysis (PCA), partial least squares discriminant analysis (PLS-DA), orthogonal partial least squares discriminant analysis (OPLS-DA) after the data treated by Pareto-scaling. PLS-DA and OPLS-DA analyses were used to obtain variable importance values (VIP). Differential metabolites were found using Student's $t$ test ($p<0.05$) and VIP (VIP$>$1.2).

## Statistical analysis

In this study, differentiation between two groups were analyzed by utilizing the Student's t-test. Spearman's correlation was performed to analyse the correlation by SPSS 18.0. A $P$ value less than 0.05 was consider as statistical significance.

# Results

## Morphology changes in response to low temperature stress

The low temperature exerted remarkably inhibition effects on the growth of bitter gourd seedlings. As shown in **Fig 1**, the damage of low temperature to the cold-susceptible bitter gourd Y17 seedlings was obvious after 2 h of 5˚C exposure. The leaves of Y17 seedlings appeared the leaves rolling and water loss. Many Y17 seedlings died when low temperature treatment was prolonged to 24 h. However, the obvious damage of low temperature on the morphology of Y54 seedlings appeared after exposing to low temperature for 8 h. After 24 h of the treatment, Y54 seedlings were still survival and but the leaves were rolled and water loss was also observed.

## MDA and $H_2O_2$ content in response to low temperature

With increasing times exposed to the low temperature, the content of MDA and $H_2O_2$ in two bitter gourds were both increased (**Fig 2**). After 24 h of low temperature treatment, MDA content of the bitter gourd Y17 and Y54 were increased 97.80% and 89.29% as compare with them before the cold treatment, respectively ($P < 0.05$). After the treatment for 8 h, 12 h and 24 h, Y54 had a lower content of MDA and $H_2O_2$ than Y17 ($P < 0.05$). The content of MDA in Y54 was 3.65±0.51 umol g$^{-1}$ at 24 h after the low temperature treatment, which was lower than Y17

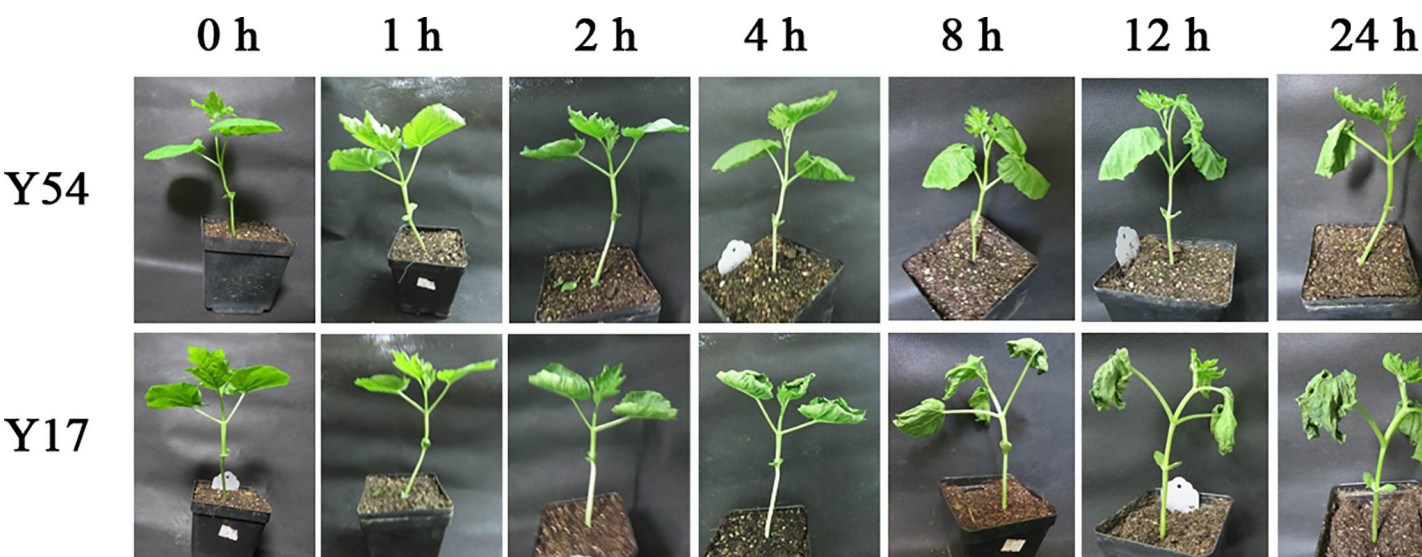

**Fig 1. Performance at the seedling stage of cold-susceptible bitter gourd Y17 and cold-resistant bitter gourd Y54 after low temperature treatment.**

by 19.41%. Maximum contents of $H_2O_2$ (2.48±0.074 mmol $g^{-1}$) were observed in the Y17 after 24 h cold treatment, which was also higher than Y54 ($P < 0.05$).

## Changes in enzymatic activities, proline content, $O_2^-$ and ·OH in response to low temperature

As shown in Fig 3A–3D, the activities of POD, SOD and CAT and the content of proline content of in the bitter gourd seedlings were evaluated at different time points after low temperature treatment. With the increased time exposed to low temperature, four indexes of bitter gourd Y17 and Y54 were exerted as an increasing trend, especially at 8 h after treatment. At 24

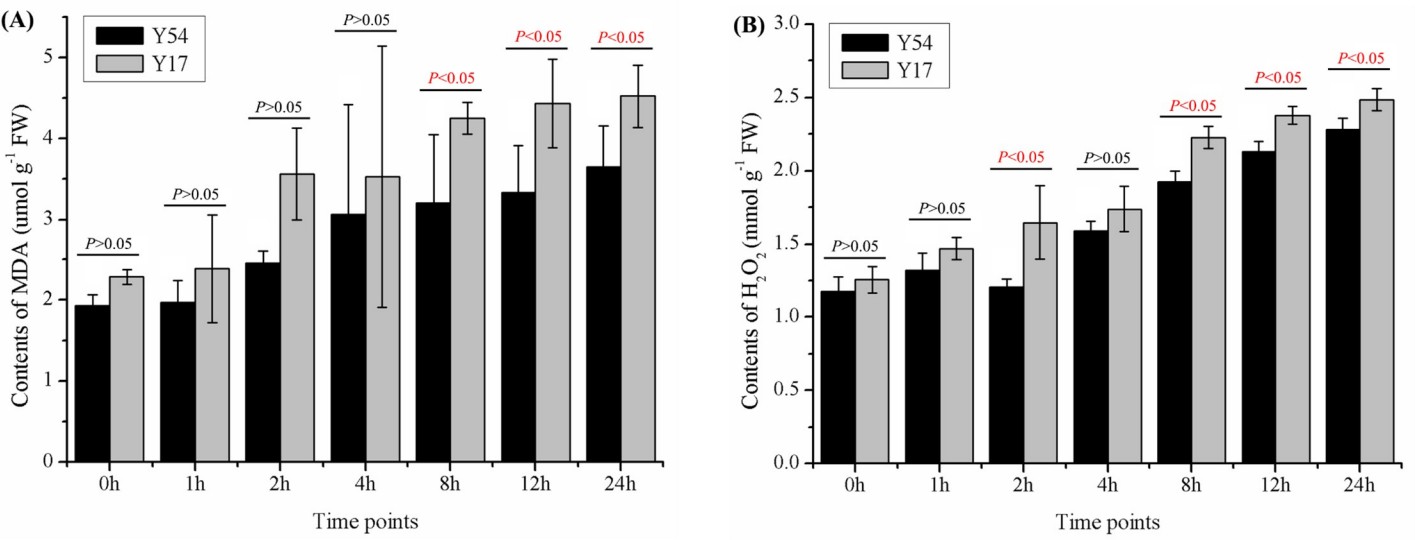

**Fig 2. Effects of cold stress on MDA and $H_2O_2$ content in bitter gourd seedlings at different time points.** The data represent means ± SE. (A) MDA contents in bitter gourd seedlings Y17 and Y54; (B)$H_2O_2$ contents in bitter gourd seedlings Y17 and Y54. $P$ value less than 0.05 exerted significance between Y17 and Y54 at in the same timepoint after cold treatment. $P$ value more than 0.05 exerted no significance.

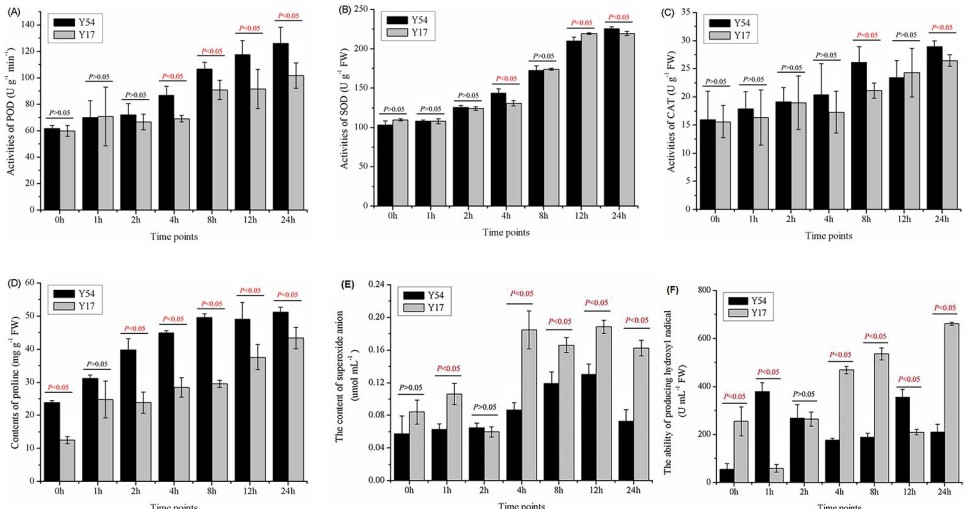

**Fig 3. Effect of cold stress on POD, SOD, CAT, proline, ·OH and $O^{2}$- in bitter gourd seedlings at different timepoints.** (A) POD activities in bitter gourd seedlings Y17 and Y54; (B) SOD activities in bitter gourd seedlings Y17 and Y54; (C) CAT activities in bitter gourd seedlings Y17 and Y54; (D) Proline contents in bitter gourd seedlings Y17 and Y54; (E); $O^{2}$- contents in bitter gourd seedlings Y17 and Y54 (F) ·OH contents in bitter gourd seedlings Y17 and Y54. The data represent means ± SE. *P* value less than 0.05 exerted significance between Y17 and Y54 at in the same timepoint after cold treatment. *P* value more than 0.05 exerted no significance.

h after the cold treatment, POD, SOD and CAT activity of Y17 were 101.67±9.60 U $g^{-1}min^{-1}$, 219.48±2.52 U $g^{-1}$ and 26.47±0.017 U $g^{-1}$, respectably. They were significantly higher than that before the treatment by 69.44%, 100.02% and 69.61%, respectively ($P < 0.05$).

By comparison with Y17, Y54 presented significantly higher activities of POD, SOD and CAT and the content of proline after exposing to low temperature. The POD activity of Y54 was found to be 61.67±2.45 U $g^{-1}min^{-1}$ at the beginning of exposure to low temperature, while it increased by 40.54% after 4 h, 72.97% after 8 h and 104.32% after 24 h of 5°C exposure, respectively (**Fig 3A**). The POD activity of the bitter gourd Y54 was significantly higher than Y17 by 23.93% after 24 h low temperature treatment ($P < 0.05$). The SOD activity of Y54 at 5°C was found to be 103.09±5.17 U $g^{-1}$ at beginning of the treatment, but this activity was increased to 225.64±2.31 U $g^{-1}$ after 24 h cold treatment (**Fig 3B**). As compared to the beginning treatment, CAT activity of Y54 after 24 h treatment was elevated by 81.73%, and which was also higher than Y17 by 9.25% (**Fig 3C**). After growing at 5°C for 24 h, the proline content of Y54 was 51.16±1.58 mg $g^{-1}$, and it was increased by 114.32% as to compare with them at 0 h. The proline content of Y54at 24h was also higher than Y17 by 17.85% (**Fig 3D**). Meanwhile, the expression of $O_2$- was dramatically increased as cold treatment longer, which was observed both in bitter gourd seedlings Y17 and Y54 (**Fig 3E**). Notably, the production in cold susceptible species Y17 was nearly two-folds than cold resistant ones Y54 at 24 hours post cold treatment. At the same time, the production of ·OH was stable in Y54, but dramatically increased in Y17 as cold treatment longer (**Fig 3F**). As an expect, the production of ·OH in Y54 was higher than Y17 at 1, 12 hours post cold treatment.

The correlation analysis of POD, SOD, CAT, $H_2O_2$, proline, $O_2$- and ·OH was also conducted. It was found that MDA of the bitter gourd showed a significant correlation with POD, SOD, CAT, $H_2O_2$, $O_2$- and ·OH content. But no significant correlation was found between MDA content and proline content (**Table 1**). The production of proline showed a significant correlation with POD, SOD, CAT and $H_2O_2$. Antioxidant enzymes (POD, SOD, CAT) showed a significant correlation with $H_2O_2$.

**Table 1. The relationship of physiological indexes (Y54, 24h post cold treatment).**

|  | Proline | MDA | SOD | POD | CAT | $H_2O_2$ | $O_2-$ | ·OH |
|---|---|---|---|---|---|---|---|---|
| Proline | 1 |  |  |  |  |  |  |  |
| MDA | 0.40 | 1 |  |  |  |  |  |  |
| SOD | 0.73** | 0.81** | 1 |  |  |  |  |  |
| POD | 0.87** | 0.57* | 0.90** | 1 |  |  |  |  |
| CAT | 0.84* | 0.68** | 0.92** | 0.93** | 1 |  |  |  |
| $H_2O_2$ | 0.57* | 0.92** | 0.94** | 0.80** | 0.84** | 1 |  |  |
| $O_2-$ | 0.16 | 0.74** | 0.54* | 0.30 | 0.31 | 0.69** | 1 |  |
| ·OH | 0.11 | 0.57* | 0.36 | 0.20 | 0.26 | 0.49 | 0.54* | 1 |

* Indicates a significant difference at the 0.05 level.

** Indicates a significant difference at the 0.01 level.

## Differentially expressed genes in response to low temperature stress and validated by Q-PCR

Using the principal component analysis, two bitter gourd seedlings at three timepoints (0, 8, 24h) could be apparently separated by two groups (Fig 4A). Moreover, 46 differentially expressed genes associated with cold acclimation were selected and annotated after the bitter gourd exposed at 5˚C (Fig 4B). Among them, 69.57% (n = 32) of genes encoded transcription factors, 8.70% (n = 4) encoded peroxidase and others encoded SOD, sucrose synthase, pyruvate decarboxylase, glutamate dehydrogenase, chalcone synthase, phosphofructokinase, nerolidol synthase and germacrene D synthase. As compared to 0 h of the treatment, four POD genes in the Y54 showed 5 to 25 folds than that at 8 h after low temperature treatment, and 6 to-452 folds than at 24 h, respectively. However, three of them in the Y 17 were also showed increased gene expression level at 8 h after the treatment, and increased levels ranged from 0.28–1.05 folds. Continued low temperature treatment for 24 h led to down-regulated expression of three POD genes. SOD gene of Y17 at 8 h and 24 h after low temperature treatment showed 1.36- and 5.52-fold increased gene expression level, respectively. In Y54, gene expression level of SOD gene increased 2.35 and 11.02 folds. After 8 h low temperature treatments, there were 27 and 23 up-regulated expression of transcription factor genes in Y54 and Y17, respectively. While the numbers of up-regulated expression genes changed to 27 and 25 after 24 h of the low temperature treatment. The elevated gene expression levels that were higher in the Y54 than that in the Y17 consisted of mainly transcription factor after exposing to low temperature for 8 h and 24 h, including McERF, McMYB, McWRKY, McTCP, McRAX, McPCL, McGATA, McE2F and McbZIP (Fig 4B).

Moreover, the different expressed genes were validated by Q-PCR, including McSOD1, McSUS1, McPDC1, McCHS1, McNES1, McERF1, McERF4, McERF7, McERF11, McMYB7, McWRKY4, McUNE1, McTCP1, McGATA1 and McAt2g1(Fig 4C). Among them, McPDC1, McCHS1, McNES1, McERF1, McERF4, McERF7, McERF11, McMYB7, McWRKY4, McUNE1, McTCP1 and McGATA1 were consistent to the transcriptomics results. As an expect, McSOD1 and McSUS1, higher expression in Y17 at time of 24h post cold treatment, was declined in Q-PCR results; however, transcriptomics results of these two genes in Y54 was consistent to its Q-PCR result. McAt2g1, higher expression in Y54 at time of 0, 24h post cold treatment, was declined in Q-PCR results; however, transcriptomics results of this gene in Y17 was consistent to its Q-PCR result. With the Q-PCR validation, the core genes for cold resistance might be McSOD1, McPDC1, McCHS1, McERF7 and McUNE1.

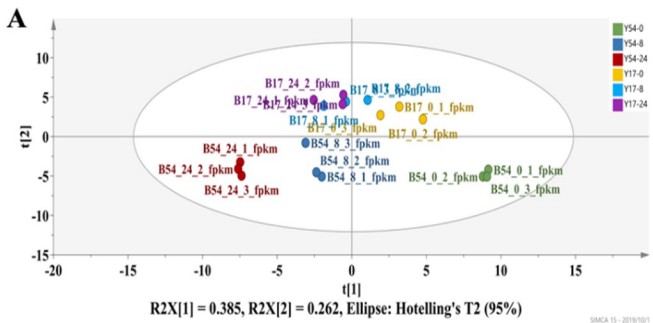

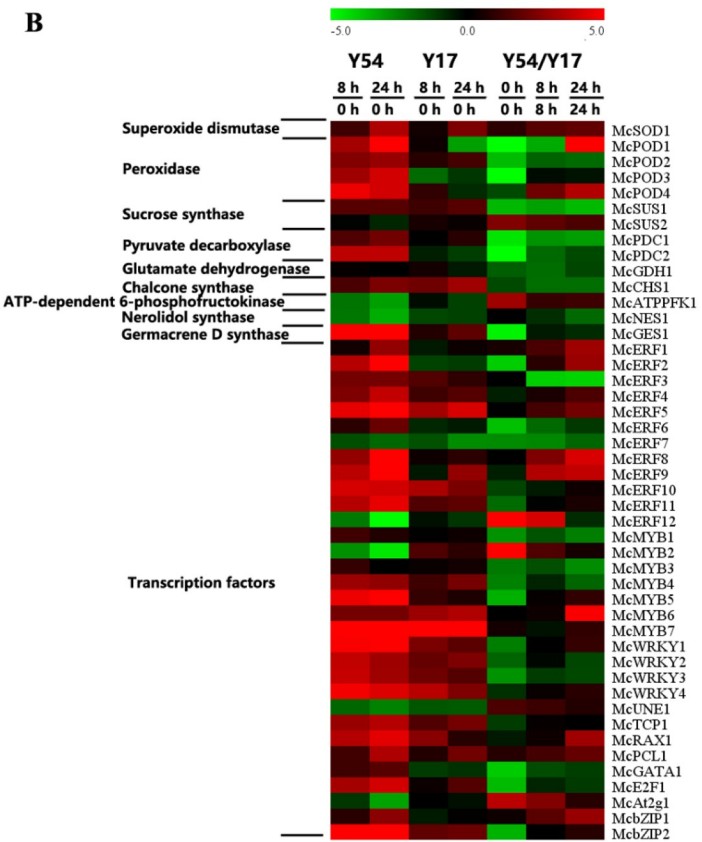

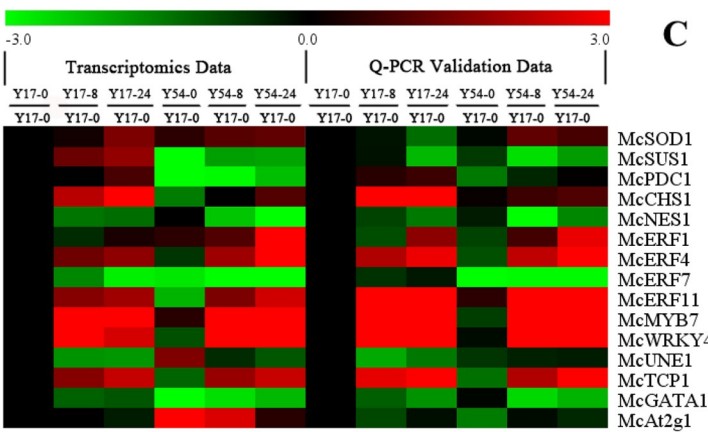

**Fig 4. Effects of low temperature on the cold induced gene expressions in leaves of bitter gourd by transcriptomics and Q-PCR validation.** (A) Principal Component Analysis results between Y17 and Y54 at in the three timepoints (0, 8, 24h) after cold treatment; (B)Heatmap of differentiated genes between Y17 and Y54 at in the three timepoints (0, 8, 24h) after cold treatment;(C) Genes validated by Q-PCR between Y17 and Y54 at in the three timepoints (0, 8, 24h) after cold treatment.

## Differentially expressed metabolites in response to low temperature stress

Based on the metabolic responses of Y54 and Y17 under low temperature stress, 195 differential expressed metabolites, including 48 amino acids, 2 polypeptides, 35 sugars, 10 fatty acids, 38 organic acids, 11 flavonoids, 7 nucleobases, 4 coenzyme, 1 steride and 39 other compounds were screened out and analyzed. Among them, 132 metabolites had greater accumulation levels in the Y54 and Y17 after 8 h and 24 h low temperature treatment, including 37 amino acids, 2 polypeptides, 33 sugars, 3 fatty acids, 22 organic acids, 1 flavonoid, 7 nucleobases, 3 coenzyme, 1 steride and 22 other compounds. As compared to the bitter gourd Y17, 115 metabolites had greater accumulation levels in Y54 after exposing to low temperature for 8 h and 24 h., including 35 amino acids, 24 sugars, 10 fatty acid, 19 organic acids, 1 flavonoid, 7 nucleobase, 2 coenzyme, 1 steride and 16 other compounds. A further comparative analysis indicated the main difference expression of sugar in the bitter gourd Y54 and Y17 were sucrose and raffinose (**Fig 5**). Moreover, these initial metabolites, including amimo acids, polypeptides, sugars, organic acids and nucleobases, were apparently increased in cold resistant species Y54 than cold susceptible species Y17, indicating that these metabolites might contribute to the cold tolerance.

## Discussion

### Damages of low temperature stress on the growth of bitter gourd seedlings and the mobilization of antioxidant mechanisms

Low temperature is a major environmental stress seriously affects plant growth, distribution and productivity [1]. Many crops originated from tropical origin are highly sensitive to low non-freezing temperatures due to their bad chilling adjustment ability [3]. Plant cells produce excess of reactive oxygen species (ROS), such as $H_2O_2$, $O_2^-$ and -OH, and the accumulation of MDA were subsequently found. The high expression of ROS could result in the deep damage of plants, such as lipid peroxidation and electrolyte leakage [20]. Notably, MDA produced in membranes could destroy membrane structure and cause protein degradation, and then it aggravated by the damage of low temperature [21] [22]. For keeping the balance of ROS numbers, plants had evolved some enzymatic and non-enzymatic antioxidant systems to eliminate or reduce the ROS damage induced by low temperature stress [1] [23]. SOD, POD and CAT were considered as the most important antioxidant enzymes to eliminate ROS. The main function of SOD is converting highly reactive $O_2^-$ to $O_2$ and $H_2O_2$. Meanwhile, POD and CAT could disintegrate produced $H_2O_2$ into water and molecular oxygen [23] [24]. This was a normal defensive response of the plant to low temperature stress, which was benefit to the plant with higher cold tolerance to survival in the colder environment or a long time of low temperature stress [25] [26].

In this work, our results also showed that low temperature exerted a serious damage to the growth of bitter gourd seedlings, such as higher expression of ROS and MDA. Meanwhile, the self-protective system of bitter gourd seedling was also restarted with the elevated activity of SOD, POD and CAT. Notably, sustained low temperature exposing even caused the death of cold-susceptible bitter gourd seedlings (**Fig 1**). Notably, as compared to cold susceptible species Y17, cold resistant ones Y54 exerted enhanced activity of SOD, POD and CAT, and less

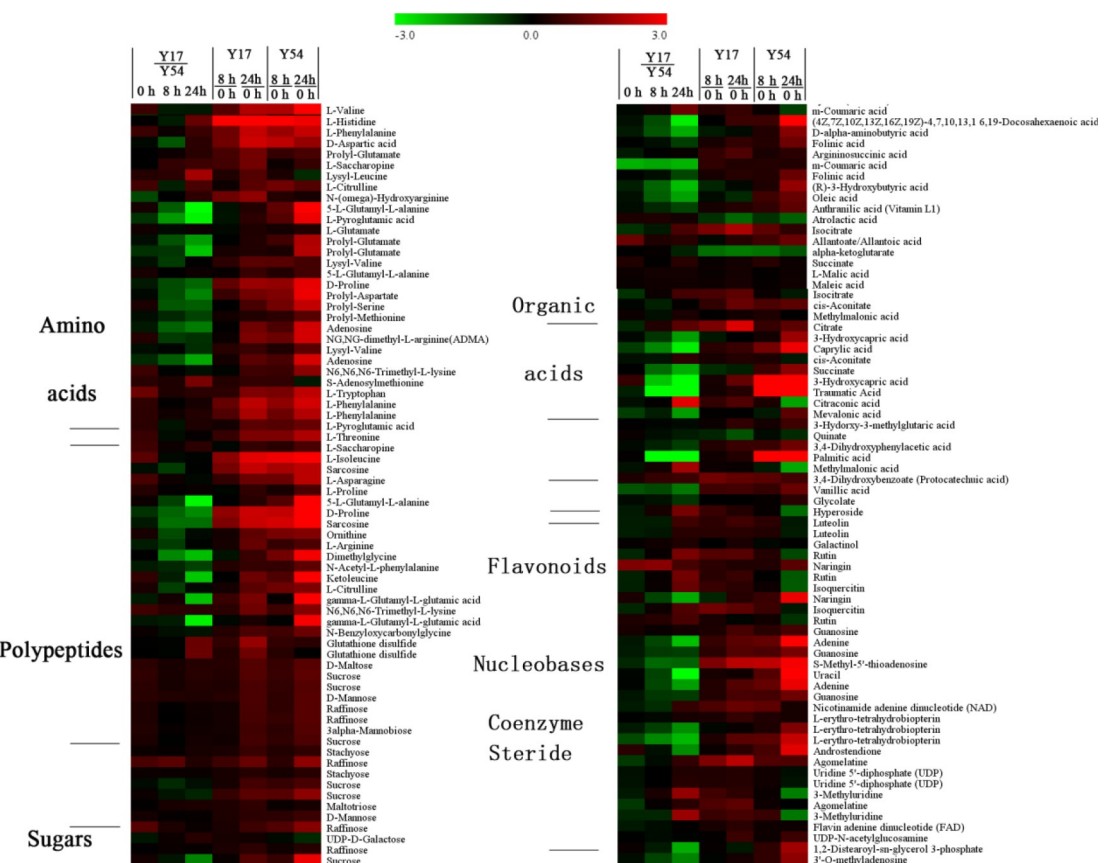

**Fig 5. Differentially expressed metabolites between bitter gourd seedlings Y17 and Y54 in response to low temperature stress by metabolomics.**

ROS and MDA were found. Improved antioxidant system as the reason why Y54 exerted higher cold tolerance.

## Cold acclimation response in bitter gourd involves transcription factors associated with cold acclimation

Transcription factor was a critical element of all gene regulatory networks that mediated various aspects of plant growth and developmental processes. It also played a central role in controlling the expression of cold-responsive genes in cold acclimation [27] [28]. The results showed that the expression levels of most transcription factor genes were up-regulated in the bitter gourd subjected to 5°C for 8 h and 24 h (Fig 4). These findings strongly suggested that the transcription factor played an important role in cold acclimation process in bitter gourd. Moreover, with the validation of Q-PCR, the core genes were found, including McSOD1, McPDC1, McCHS1, McERF7 and McUNE1. Among them, McSOD1 was correlated to the production of SOD which had high ability to eliminate the ROS in plants, such as $H_2O_2$, $O_2$-, *etc* [29]. Elevated McSOD1 was found in cold resistant bitter gourd seedling Y54. McPDC1 encoded pyruvate decarboxylase which involved in glycolytic pathway, and higher expression resulted in the low sugars and higher alcohol which damaged the growth of plant [30]. It was found that McPDC1 in Y54 was lower than Y17. Meanwhile, the sugars in Y54 was correspondingly higher than Y17. CHS1 (chilling-sensitive mutant 1), encoded by McCHS1,

displayed a chilling-sensitive phenotype, and also displayed defense-associated phenotypes, including extensive cell death, the accumulation of hydrogen peroxide and salicylic acid [31]. In our work, the higher expression of McCHS1 was found in cold susceptible Y17 and lower expression in cold resistant bitter gourd seedling Y54. McERF7 encoded ethylene-response factor 7 (ERF7) which could be downregulation after cold treatment [32]. Its expression could be regulated by plant defense inducers such as ethylene, jasmonic acid and salicylic acid, and overexpression of ERFs in transgenic plants leaded to disease resistance [33]. However, the ERF7 was downregulated in Y17 and less downregulated in Y54. However, the function of McUNE1, downregulated in Y17 and less downregulated in Y54, could be not found in plants. Based on the above information, enhanced McSOD1, downregulated of McPDC1 and less improved McCHS1 were the core anti-cold mechanism of cold resistant bitter gourd seedlings Y54.

## Cold-induced osmoregulation substance expression

In response to the low temperature stress, the plant accumulated a range of osmoregulation substance to resist the cold stress, including sugar, fatty acid and amino acid [10]. A lot of evidence had demonstrated that there was a significant positive correlation between the accumulation of proline and improved cold tolerance in the plant [3, 34]. Proline can regulate cytosol acidity and decrease lipid peroxidation to improve plant tolerance of low temperature stress [35]. In this study, there was an obvious increasing trend of the proline content in the bitter gourd after exposing to the low temperature stress (**Fig 3**). It suggested that the bitter had the ability of improving the proline contents to resist the cold stress. We also found that the glutamate which participated in synthesizing the proline was significantly improved after exposing to the low temperature stress (**Fig 5**). In addition, the cold-resistant bitter gourd Y54 could accumulate more proline than the cold-susceptible bitter gourd Y17 after 2 h treatment, which suggested that the cold-resistant bitter gourd had a faster and better chilling acclimation to cold stress.

Carbohydrate metabolism had been reported to be a main contributed process of chilling acclimation, and the precursors or intermediate products of carbohydrate compounds all could play roles as osmoregulators, cryoprotectants or signaling molecules [36] [37]. Soluble carbohydrate could regulate cell osmotic potential to maintain the balance of osmotic pressure, and protect plasma membrane during cold stress [38]. In this study, 33 sugars had up-regulated expression levels in the bitter gourd after 8 h and 24 h low temperature treatment. These up-regulations suggested that there was a tight link between carbohydrate metabolism and cold stress. The bitter gourd could adjust carbohydrate metabolism to tolerant cold stress. The accumulation of sucrose as an osmoprotectant supported the role of the plant to stabilize cellular membranes and maintains turgor, when they exposed to low temperature stress [38]. The metabolites profiling analysis showed that the sucrose had greater accumulation levels in the bitter gourd at 8 h and 24 h after low temperature treatment. Meanwhile, the expression level of sucrose synthase was up-regulated, which suggested that the sucrose metabolism also presented an important role in chilling acclimation of the biter gourd. After exposing to the low temperature stress for 8 h and 24 h, the expression level of raffinose in the bitter gourd has significantly elevated (**Fig 5**). This phenomenon of increased raffinose content also could be found in other plant subjected to cold stress, such as maize, Siberian spruce [39]. The role of the raffinose has been implicated in membrane protection and radical scavenging, and the accumulation of raffinose content is contributed to elevate cold tolerance of the plant [40]. Therefore, carbohydrate metabolism plays an important role in chilling acclimation of the bitter gourd.

## Conclusions

In this study, our result demonstrated that the low temperature stress exerted obvious damages to the growth of the chilling sensitive bitter gourd seedlings and anti-cold mechanism of cold resistant species were revealed in metabolism and gene levels. When facing cold treatment, the bitter gourd seedlings also restarted self-protective mechanism to tolerance the cold stress. This defense mechanisms included the accumulation of osmoregulation substance, such as amino acid, sugar, fatty acid and organic acid, and the mobilization of antioxidant systems, such as up-regulating expression levels of genes related cold tolerant and increasing antioxidant enzyme activities. The core genes were validated by Q-PCR, including McSOD1, McPDC1 and McCHS1, were found to exert great importance in maintain the balance between oxidants and anti-oxidants. The cold-resistant bitter gourd has a better performance than the cold-susceptible bitter gourd in these aspects. Based on the result, we expect that this work can provide new knowledge and idea to breeding cold tolerance bitter gourd.

## Author Contributions

**Conceptualization:** Ziji Liu, Zhiqiang Qi, Yan Yang.

**Data curation:** Huang He.

**Methodology:** Xu Han.

**Writing – original draft:** Yu Niu, Ziji Liu.

**Writing – review & editing:** Zhiqiang Qi, Yan Yang.

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
