## [Decision Letter · Decision Letter 0]

25 Feb 2020

PONE-D-19-35964

Gene expression and metabolic changes of Momordica Charantia L. seedlings in response to low temperature stress

PLOS ONE

Dear Dr Niu,

Thank you for submitting your manuscript to PLOS ONE. After careful consideration, we feel that it has merit but does not fully meet PLOS ONE’s publication criteria as it currently stands. Therefore, we invite you to submit a revised version of the manuscript that addresses the points raised during the review process.

We would appreciate receiving your revised manuscript by Apr 10 2020 11:59PM. To enhance the reproducibility of your results, we recommend that if applicable you deposit your laboratory protocols in protocols.io, where a protocol can be assigned its own identifier (DOI) such that it can be cited independently in the future. For instructions see: http://journals.plos.org/plosone/s/submission-guidelines#loc-laboratory-protocols

We look forward to receiving your revised manuscript.

Kind regards,

Haitao Shi

Academic Editor

PLOS ONE

Journal Requirements:

3. Please amend the manuscript submission data (via Edit Submission) to include author Yan Yang.

"This work was financially supported by Key Research and Development Projects of Hainan Province（ZDYF2019103）and National Nonprofit Institute Research Grant of CATAS-TCGRI (1630032017027)."

"he author(s) received no specific funding for this work."

Reviewers' comments:

Reviewer's Responses to Questions

**Comments to the Author**

1. Is the manuscript technically sound, and do the data support the conclusions?

Reviewer #1: No

Reviewer #2: Yes

Reviewer #3: Partly

2. Has the statistical analysis been performed appropriately and rigorously? 

Reviewer #1: No

Reviewer #2: N/A

Reviewer #3: Yes

3. Have the authors made all data underlying the findings in their manuscript fully available?

Reviewer #1: No

Reviewer #2: Yes

Reviewer #3: Yes

4. Is the manuscript presented in an intelligible fashion and written in standard English?

Reviewer #1: No

Reviewer #2: Yes

Reviewer #3: No

5. Review Comments to the Author

Reviewer #1: The manuscript entitled: “Gene expression and metabolic changes of Momordica Charantia L. seedlings in response to low temperature stress” stated a work on the effects of low temperature on the growth, gene expression and metabolic changes in the seedlings Momordica charantia L. The authors conclude that the cold tolerance mechanisms in Momordica charantia L. are related to the mobilization of antioxidant systems, the adjustment of the transcription factors and accumulation of osmoregulation substance. The overall quality of this manuscript is poor, so it requires much more work to be done before publication. Here below are some major concerns and questions:

1. The manuscript gives me a general impression that it is just an experimental report. The preparation and writing of manuscripts are very rough.

-- the title: “charantia”, “c” should be lower case letter

-- Latin scientific name of bitter gourd should be given

-- Reference 2 is a report about Anabasis aphylla. But it was used to describe bitter gourd? Reference is a report about bitter gourd. But tomato was mentioned? Many similar situations. Please check references.

-- the 3rd paragraph, “the Zhao et al. (2019) showed that”, “Barrero-Gil et al. (2016) found that”, reference citation format is different from others.

-- Reference 13: Northern Horticulture, Reference 28: American Society for Horticultural Sciencewhat’s Journal?

2. Although, I think that authors need to specify the aim of the study, it is not the art to show all the data that are done in the lab, they should be logically linked, but I think the authors missed this aspect. For example, what is the explanation for examination of effects of cold stress on gene expression and metabolites? What is the relationship between differentially expressed metabolites and differentially expressed genes in response to low temperature stress? It is not specified in the context of the whole work.

3. Materials and Methods

--I have a question, it seems that there no the control treatment in the experiment design. There are 7 time points (0, 1, 2, 4, 8, 12 and 24 h). I think the normal temperature treatment which does not cause cold stress should be included to clarify the objective.

4. Moreover, there is no logical and sound discussion of the topic, but a merely list of results achieved from original experiments. A deepgoing discussion should be considered. What is the mechanism of the effects of cold stress on differentially expressed genes? How can cold stress regulate differentially expressed metabolites?

5. Figures

-- the lower case letters used in the Figures should be clearly stated.

-- Figure 2A: the error bar is so high, and the results should be reconsidered.

-- Figure 3B: 12 h, a and b, are the positions swapped? This made me very confusing.

6. The conclusion part is exaggerated. For example, “In addition, chilling acclimation was regulated by a similar set of transcription factors, including bZIP, MYB and WRKY family.” Usually, the conclusion should be based on your results.

7. I have, however, difficulty in reading the manuscript mainly because of its premature language usage. Some places exist formatting, grammar and other mistakes. It needs an improvement, if possible, I request the authors to have the manuscript English edited by a professional language editor for scientific writing.

Reviewer #2: In this study, authors investigated the cold stress response in bitter gourd at phenotypic,

physiological and metabolic levels. These results suggested that the bitter gourd exhibited higher cold tolerance may attribute to the mobilization of antioxidant systems, the adjustment of the transcription factors and accumulation of osmoregulation substance. Compared with the cold-susceptible bitter gourd, the cold-resistant bitter gourd has a better performance in these aspects.This research might provide useful information leading to how bitter gourd can adapt to the low temperature stress. However, the manuscript need to be modified based on the comments below for the publication in this journal.

1. Please increase the line number

2.Introduction：The references about cold injury are too general, please add some references from bitter gourd cultivars, herbaceous plants or vegetables, The adaptation mechanism of woody and herbaceous plants to cold injury is quite different.

3. Materials and methods:What is the basis for choosing 5 ℃ low temperature treatment?

Increase more information about materials cold-susceptible (Y17) and cold-resistant (Y54)), especially for temperature tolerance.

4.The authors assayed for change in malondialdehyde (MDA), hydrogen peroxide (H2O2), proline content and activities of antioxidant enzymes. this work provided a large amount of data, which might be informative for some potential readers.Majority of guaiacol peroxidase activity is derived from extracellular peroxidases, which are probably involved in many biological processes (such as lignin and suberin formation, cross-linking of cell wall components, and/or synthesis of phytoalexins), and non-specific peroxidases located in the vacuole. Their contribution to antioxidative defense is largely unclear. Ascorbate peroxidases and thiol-dependent peroxidases (such as peroxiredoxin), which are the representative antioxidative enzymes in plants, are much better for your study. Additional experiments, such as checking O2- and/or ROS levels, should be added.

5. In this paper, the author sequenced the transcriptome of the two groups of samples, but there was no verification of the difference gene. Additional experiments, such as validation by qRT-PCR analysis, should be added.

Reviewer #3: 1. Is the manuscript technically sound, and do the data support the conclusions?

Some data are not clear, such as in Table 1, genotype and time information should be included, and should be representative. Field performances of the Y17 and Y54 in low temperature stress conditions should be given.

4. Is the manuscript presented in an intelligible fashion and written in standard English?

The English text should be greatly improved.

6. PLOS authors have the option to publish the peer review history of their article (what does this mean?). If published, this will include your full peer review and any attached files.

Reviewer #1: No

Reviewer #2: No

Reviewer #3: No

---

## [Author Response · Author response to Decision Letter 0]

28 Apr 2020

Dear Editors:

Thank you for your comments about our manuscript entitled “Gene expression and metabolic changes of Momordica charantia L. seedlings in response to low temperature stress ” (ID: PONE-D-19-35964). Those comments are all valuable and helpful to improve our work. We have studied comments carefully and revised our manuscript. The revised parts are marked in red. The responds for the reviewer’s comments are listed as follows:

Comments from reviewers:

Reviewer #1

The manuscript entitled: “Gene expression and metabolic changes of Momordica Charantia L. seedlings in response to low temperature stress” stated a work on the effects of low temperature on the growth, gene expression and metabolic changes in the seedlings Momordica charantia L. The authors conclude that the cold tolerance mechanisms in Momordica charantia L. are related to the mobilization of antioxidant systems, the adjustment of the transcription factors and accumulation of osmoregulation substance. The overall quality of this manuscript is poor, so it requires much more work to be done before publication. Here below are some major concerns and questions:

1. The manuscript gives me a general impression that it is just an experimental report. The preparation and writing of manuscripts are very rough.

-- the title: “charantia”, “c” should be lower case letter

Response: We are sorry for the mistake and correction was made in manuscript. Meanwhile, the manuscript was re-edited by a scholar who is the expert in English. 

-- Latin scientific name of bitter gourd should be given.

Response: Thanks for your suggestion, the latin scientific name of bitter gourd as “Momordica Charantia L.” was provide in manuscript marked in red.

-- Reference 2 is a report about Anabasis aphylla. But it was used to describe bitter gourd? Reference is a report about bitter gourd. But tomato was mentioned? Many similar situations. Please check references.

Response: Thanks for your suggestion. We used the reference 2 to emphasize the impact of low temperature on the germination, growth and distribution of the plant so as tomato. Some detected method in reference 2 was also used in our work. Meanwhile, we are sorry for the confusion of references and the correction was made in manuscript.

-- the 3rd paragraph, “the Zhao et al. (2019) showed that”, “Barrero-Gil et al. (2016) found that”, reference citation format is different from others.

Response: We are sorry for the mistake and correction of references was made one by one in manuscript. “the Zhao et al. (2019) showed that” was corrected as “By using the transcriptome method, researcher found that” and reference was added (reference 17). “Barrero-Gil et al. (2016) found that” was corrected as “Meanwhile, researcher, using the transcriptome analysis, found that” and reference was added (reference 3).

Reference: 3. Barrero-Gil J, Huertas R, Luis Rambla J, Granell A, Salinas J. Tomato plants increase their tolerance to low temperature in a chilling acclimation process entailing comprehensive transcriptional and metabolic adjustments. Plant Cell and Environment. 2016;39(10):2303-18. doi: 10.1111/pce.12799. PubMed PMID: WOS:000385846100015.

17. Zhao X, Wei J, He L, Zhang Y, Zhao Y, Xu X, et al. Identification of Fatty Acid Desaturases in Maize and Their Differential Responses to Low and High Temperature. Genes. 2019;10(6). doi: 10.3390/genes10060445. PubMed PMID: WOS:000473797000040.

-- Reference 13: Northern Horticulture, Reference 28: American Society for Horticultural Sciencewhat’s Journal?

Response: We are sorry for the confusion and correction was made in manuscript marked in red. The journal name of reference 13 was right since it was a Chinese journal named “北方园艺” and corrected in revised manuscript as reference 14. The journal name of reference 28 was deleted since the discussion was revised.

2. Although, I think that authors need to specify the aim of the study, it is not the art to show all the data that are done in the lab, they should be logically linked, but I think the authors missed this aspect. For example, what is the explanation for examination of effects of cold stress on gene expression and metabolites? What is the relationship between differentially expressed metabolites and differentially expressed genes in response to low temperature stress? It is not specified in the context of the whole work.

Response: Thanks for your suggestion. The linkage between cold temperature and plant metabolites was discussed in the “Discussion” section marked in red. such as “Plant cells produce excess of reactive oxygen species (ROS), such as H2O2, O2- and -OH, and the accumulation of MDA were subsequently found. The high expression of ROS could result in the deep damage of plants, such as lipid peroxidation and electrolyte leakage [20]”, “In this work, our results also showed that low temperature exerted a serious damage to the growth of bitter gourd seedlings, such as higher expression of ROS and MDA. Meanwhile, the self-protective system of bitter gourd seedling was also restarted with the elevated activity of SOD, POD and CAT. Notably, sustained low temperature exposing even caused the death of cold-susceptible bitter gourd seedlings (Figure 1). Notably, as compared to cold susceptible species Y17, cold resistant ones Y54 exerted enhanced activity of SOD, POD and CAT, and less ROS and MDA were found. Improved antioxidant system as the reason why Y54 exerted higher cold tolerance.” Meanwhile, the detailed data was deleted.

Added reference:20. Mishra V, Srivastava G, Prasad SM. Antioxidant response of bitter gourd (Momordica charantia L.) seedlings to interactive effect of dimethoate and UV-B irradiation. Scientia Horticulturae. 2009;120(3):373-8. doi: 10.1016/j.scienta.2008.11.024. PubMed PMID: WOS:000264648400012.

3. Materials and Methods

--I have a question, it seems that there no the control treatment in the experiment design. There are 7 time points (0, 1, 2, 4, 8, 12 and 24 h). I think the normal temperature treatment which does not cause cold stress should be included to clarify the objective.

Response: We are so sorry for our confused expression. Actually, in our work, the timepoint of 0h was the control treatment in our experiment design. The changes among them were also discussed in our work. 

4. Moreover, there is no logical and sound discussion of the topic, but a merely list of results achieved from original experiments. A deepgoing discussion should be considered. What is the mechanism of the effects of cold stress on differentially expressed genes? How can cold stress regulate differentially expressed metabolites?

Response: Thanks for your suggestion. In our manuscript, the deep discussion was added in manuscript marked in red as “Moreover, with the validation of Q-PCR, the core genes were found, including McSOD1, McPDC1, McCHS1, McERF7 and McUNE1. Among them, McSOD1 was correlated to the production of SOD which had high ability to eliminate the ROS in plants, such as H2O2, O2-, etc [29]. Elevated McSOD1 was found in cold resistant bitter gourd seedling Y54. McPDC1 encoded pyruvate decarboxylase which involved in glycolytic pathway, and higher expression resulted in the low sugars and higher alcohol which damaged the growth of plant [30]. It was found that McPDC1 in Y54 was lower than Y17. Meanwhile, the sugars in Y54 was correspondingly higher than Y17. CHS1 (chilling-sensitive mutant 1), encoded by McCHS1, displayed a chilling-sensitive phenotype, and also displayed defense-associated phenotypes, including extensive cell death, the accumulation of hydrogen peroxide and salicylic acid [31]. In our work, the higher expression of McCHS1 was found in cold susceptible Y17 and lower expression in cold resistant bitter gourd seedling Y54. McERF7 encoded ethylene-response factor 7 (ERF7) which could be downregulation after cold treatment [32]. Its expression could be regulated by plant defense inducers such as ethylene, jasmonic acid and salicylic acid, and overexpression of ERFs in transgenic plants leaded to disease resistance [33]. However, the ERF7 was downregulated in Y17 and less downregulated in Y54. However, the function of McUNE1, downregulated in Y17 and less downregulated in Y54, could be not found in plants. Based on the above information, enhanced McSOD1, downregulated of McPDC1 and less improved McCHS1 were the core anti-cold mechanism of cold resistant bitter gourd seedlings Y54.”.

Meanwhile, the excessive discussion part was deleted, including “These differentially expressed genes associated with cold acclimation encoded transcription factors could also find in other cold tolerant plant [3] [32]. bZIPs transcription factor had potential ability to activate the expression of ProDH gene, which participated in the metabolism of proline in Arabidopsis plants [33]. In this study, the expression levels of McbZIP1 and McbZIP2 transcription factors of the bitter gourd Y54 were up-regulated after exposing to the low temperature (Figure 4). It suggested that McbZIPs transcription factor participated in the response of the bitter gourd to low temperature stress. WRKY transcription factor genes had been known as important roles in the regulation of transcriptional reprogramming associated with plant stress responses. For example, transgenic Arabidopsis plants overexpressing GmWRKY21 showed increased tolerance to cold stress when compared with wild-type plants [34]. This study found that four WRKY transcription factor genes were all elevated the transcription expression level after exposing to the low temperature. Therefore, it indicated that the WRKY transcription factors exerted an important role in promoting the bitter gourd to resist low temperature stress. Some researches had demonstrated that the family of MYB transcription factor involved in the expression of ABA-dependent gene and CBF genes, and then influence the performance of the plant to resist low temperature stress [35]. Six MYB transcription factor gene up-regulated expression levels after 8 h and 24 h of the low temperature treatment, which suggested that these transcript factor had a close tine to the cold tolerance mechanism in the bitter gourd.”.

Added references: 29. Anand A, Kumari A, Thakur M, Koul A. Hydrogen peroxide signaling integrates with phytohormones during the germination of magnetoprimed tomato seeds. Sci Rep. 2019;9(1):8814. Epub 2019/06/21. doi: 10.1038/s41598-019-45102-5. PubMed PMID: 31217440.

30. Andrews DL, MacAlpine DM, Johnson JR, Kelley PM, Cobb BG, Drew MC. Differential induction of mRNAs for the glycolytic and ethanolic fermentative pathways by hypoxia and anoxia in maize seedlings. Plant Physiol. 1994;106(4):1575-82. Epub 1994/12/01. PubMed PMID: 7846162; PubMed Central PMCID: PMCPMC159700.

31. Wang Y, Zhang Y, Wang Z, Zhang X, Yang S. A missense mutation in CHS1, a TIR-NB protein, induces chilling sensitivity in Arabidopsis. Plant J. 2013;75(4):553-65. Epub 2013/05/09. doi: 10.1111/tpj.12232. PubMed PMID: 23651299.

32. Zhai Y, Wang Y, Li Y, Lei T, Yan F, Su L, et al. Isolation and molecular characterization of GmERF7, a soybean ethylene-response factor that increases salt stress tolerance in tobacco. Gene. 2013;513(1):174-83. Epub 2012/11/01. doi: 10.1016/j.gene.2012.10.018. PubMed PMID: 23111158.

33. Vallejo-Reyna MA, Santamaria JM, Rodriguez-Zapata LC, Herrera-Valencia VA, Peraza-Echeverria S. Identification of novel ERF transcription factor genes in papaya and analysis of their expression in different tissues and in response to the plant defense inducer benzothiadiazole (BTH). Physiological and Molecular Plant Pathology. 2015;91:141-51. doi: 10.1016/j.pmpp.2015.06.005. PubMed PMID: WOS:000361860900018.

5. Figures

-- the lower case letters used in the Figures should be clearly stated.

-- Figure 2A: the error bar is so high, and the results should be reconsidered.

-- Figure 3B: 12 h, a and b, are the positions swapped? This made me very confusing.

Response: We are so sorry for our confused expression. the results in Figure 2A were conducted by three repeated work; its high error bar might be caused by the individual differences of plant. According to your suggestion, the significance was clearly expressed by using “P<0.05’ or “P>0.05” in all figures. 

6. The conclusion part is exaggerated. For example, “In addition, chilling acclimation was regulated by a similar set of transcription factors, including bZIP, MYB and WRKY family.” Usually, the conclusion should be based on your results.

Response: Thanks for your suggestion. Exaggerated part in our manuscript, “In addition, chilling acclimation was regulated by a similar set of transcription factors, including bZIP, MYB and WRKY family.”, was deleted and validation results were added in revised manuscript marked in red as “The core genes were validated by Q-PCR, including McSOD1, McPDC1 and McCHS1, were found to exert great importance in maintain the balance between oxidants and anti-oxidants”.

7. I have, however, difficulty in reading the manuscript mainly because of its premature language usage. Some places exist formatting, grammar and other mistakes. It needs an improvement, if possible, I request the authors to have the manuscript English edited by a professional language editor for scientific writing.

Response: Thanks for your suggestion. Our manuscript was revised by a language scholar.

Reviewer #2: 

In this study, authors investigated the cold stress response in bitter gourd at phenotypic, physiological and metabolic levels. These results suggested that the bitter gourd exhibited higher cold tolerance may attribute to the mobilization of antioxidant systems, the adjustment of the transcription factors and accumulation of osmoregulation substance. Compared with the cold-susceptible bitter gourd, the cold-resistant bitter gourd has a better performance in these aspects.This research might provide useful information leading to how bitter gourd can adapt to the low temperature stress. However, the manuscript need to be modified based on the comments below for the publication in this journal.

1. Please increase the line number

Response: Thanks for your suggestion. Line number was provided in revised manuscript.

2.Introduction：The references about cold injury are too general, please add some references from bitter gourd cultivars, herbaceous plants or vegetables, the adaptation mechanism of woody and herbaceous plants to cold injury is quite different.

Response: Thanks for your suggestion. The specify of references was provided in manuscript marked in red. “Low temperature stress induces large accumulation of free radical, destroys the dynamic equilibrium of activate oxygen metabolism inside plant bodies, and poison cells” in introduction was changed to “Low temperature stress induces enhanced accumulation of free radical, destroys the dynamic equilibrium of activate oxygen metabolism, and decreased osmotic solutes in plants”, and references were added.

Reference:

7. Wang J, Shang S, Tian L, Zhou M, Pan Q, Zou K, et al. Effects of Low Temperature Stress on Antioxidant System of Grafted Bitter Gourd Seedlings. Chinese Journal of Tropical Crops. 2018;39(2):237-45. PubMed PMID: CSCD:6232437.

8. Zou K, Shang S, Tian L, Zhu G, Zhou M, Pan Q, et al. Effects of Low Temperature Stress on Osmotic Solutes of Grafted Bitter Gourd Seedlings. Chinese Journal of Tropical Crops. 2018;39(8):1533-9. PubMed PMID: CSCD:6349447.

3. Materials and methods: What is the basis for choosing 5 ℃ low temperature treatment? Increase more information about materials cold-susceptible (Y17) and cold-resistant (Y54), especially for temperature tolerance.

Response: Thanks for your suggestion. The reason we choose the condition of 5 ℃ was that we had done the pre-experiment to screen out the cold-susceptible (Y17) and cold-resistant (Y54). It was found that six bitter gourd could be divided into three cold tolerance grades under 5 ℃ for 1 day, and the maximum cold damage index was ranged from 20.31 to 84.38. The temperature of 5 ℃ could be used as the temperature index for identification of cold tolerant species. As your suggestion, we added this information in section “Plant materials and growth conditions” as “In our pre-experiment, six bitter gourds could be divided into three cold tolerance grades under 5 ℃ for 1 day, and the maximum cold damage index was ranged from 20.31 to 84.38. Thus, the temperature of 5 ℃ could be used as the temperature index for identification of cold tolerant species.”.

4.The authors assayed for change in malondialdehyde (MDA), hydrogen peroxide (H2O2), proline content and activities of antioxidant enzymes. this work provided a large amount of data, which might be informative for some potential readers. Majority of guaiacol peroxidase activity is derived from extracellular peroxidases, which are probably involved in many biological processes (such as lignin and suberin formation, cross-linking of cell wall components, and/or synthesis of phytoalexins), and non-specific peroxidases located in the vacuole. Their contribution to antioxidative defense is largely unclear. Ascorbate peroxidases and thiol-dependent peroxidases (such as peroxiredoxin), which are the representative antioxidative enzymes in plants, are much better for your study. Additional experiments, such as checking O2- and/or ROS levels, should be added.

Response: Thanks for your suggestion, the O2- and ·OH levels were added in Figure 3E and 3F, and the detection information was added in section “Determination of physiological indices at different time interval” as “Meanwhile, superoxide anion kits (Solarbio Life Scince Co., Ltd., China) and hydroxyl radical kits (Nanjing Jiancheng Technology Co., Ltd., China) were used to detected O2- and ·OH according to the manufacturer’s guidelines.”. Moreover, results were also described in section “Changes in enzymatic activities, proline content, O2- and ·OH in response to low temperature” as “Meanwhile, the expression of O2- was dramatically increased as cold treatment longer, which was observed both in bitter gourd seedlings Y17 and Y54 (Figure 3E). Notably, the production in cold susceptible species Y17 was nearly two-folds than cold resistant ones Y54 at 24 hours post cold treatment. At the same time, the production of ·OH was stable in Y54, but dramatically increased in Y17 as cold treatment longer (Figure 3F). As an expect, the production of ·OH in Y54 was higher than Y17 at 1, 12 hours post cold treatment.”.

Figure 3E O2- contents in bitter gourd seedlings Y17 and Y54. The data represent means ± SE. P value less than 0.05 exerted significance between Y17 and Y54 at in the same timepoint after cold treatment. P value more than 0.05 exerted no significance. 

Figure 3F ·OH contents in bitter gourd seedlings Y17 and Y54. The data represent means ± SE. P value less than 0.05 exerted significance between Y17 and Y54 at in the same timepoint after cold treatment. P value more than 0.05 exerted no significance.

5. In this paper, the author sequenced the transcriptome of the two groups of samples, but there was no verification of the difference gene. Additional experiments, such as validation by qRT-PCR analysis, should be added.

Response: Thanks for your suggestion. As your suggestion, Q-PCR data was detected and provided in revised manuscript. Meanwhile, related description and discussion were also added in section “Differentially expressed genes in response to low temperature stress and validated by Q-PCR” as “Moreover, the different expressed genes were validated by Q-PCR, including McSOD1, McSUS1, McPDC1, McCHS1, McNES1, McERF1, McERF4, McERF7, McERF11, McMYB7, McWRKY4, McUNE1, McTCP1, McGATA1 and McAt2g1(Figure 4C). Among them, McPDC1, McCHS1, McNES1, McERF1, McERF4, McERF7, McERF11, McMYB7, McWRKY4, McUNE1, McTCP1 and McGATA1 were consistent to the transcriptomics results. As an expect, McSOD1 and McSUS1, higher expression in Y17 at time of 24h post cold treatment, was declined in Q-PCR results; however, transcriptomics results of these two genes in Y54 was consistent to its Q-PCR result. McAt2g1, higher expression in Y54 at time of 0, 24h post cold treatment, was declined in Q-PCR results; however, transcriptomics results of this gene in Y17 was consistent to its Q-PCR result. With the Q-PCR validation, the core genes for cold resistance might be McSOD1, McPDC1, McCHS1, McERF7 and McUNE1.

”.

Figure 4C Genes validated by Q-PCR between Y17 and Y54 at in the three timepoints (0, 8, 24h) after cold treatment.

Reviewer #3: 

1. Is the manuscript technically sound, and do the data support the conclusions?

Response: Thanks for your suggestion. Our work was re-edited and related data was collected to support our conclusion. Meanwhile, the aim of our work was to revealing the core genes and metabolites which related to bitter gourd seedling facing low temperature. And core genes were validated by Q-PCR, including McSOD1 and McERF1. Moreover, the initial metabolites, including amimo acids, polypeptides, sugars, organic acids and nucleobases, were apparently increased in cold resistant species Y54 than cold susceptible species Y17, indicating that these metabolites might contribute to the cold tolerance. 

2 Some data are not clear, such as in Table 1, genotype and time information should be included, and should be representative. 

Response: Thanks for your suggestion. The aim of Table 1 was finding the correlation relationship among the detected index. The table was re-checked and done by using the data of Y54 at 24h post cold treatment.

Table 1 The relationship of physiological indexes (Y54, 24h post cold treatment)

 Proline MDA SOD POD CAT H2O2 O2- ·OH

Proline 1 

MDA 0.40 1 

SOD 0.73** 0.81** 1 

POD 0.87** 0.57* 0.90** 1 

CAT 0.84* 0.68** 0.92** 0.93** 1 

H2O2 0.57* 0.92** 0.94** 0.80** 0.84** 1 

O2- 0.16 0.74** 0.54* 0.30 0.31 0.69** 1 

·OH 0.11 0.57* 0.36 0.20 0.26 0.49 0.54* 1

* Indicates a significant difference at the 0.05 level. ** Indicates a significant difference at the 0.01 level.

3 Field performances of the Y17 and Y54 in low temperature stress conditions should be given.

Response: Thanks for your suggestion. Detailed field performance information was provided in section “Plant materials and growth conditions” marked in red, including “The experiment was carried out in 8th Solar Greenhouses of the melon and vegetable research laboratory of Tropical Crops Genetic Resources Research Institute (Danzhou, China). The bitter gourd seeds, with the same size, were subjected to hot-water treatment and cup seedling (8cm*8cm)”.

4. Is the manuscript presented in an intelligible fashion and written in standard English? The English text should be greatly improved.

Response: Thanks for your suggestion. The expression was dramatically improved by a scholar who is expert in English.

---

## [Editor Report · Decision Letter 1]

29 Apr 2020

Gene expression and metabolic changes of Momordica c harantia L. seedlings in response to low temperature stress

PONE-D-19-35964R1

Dear Dr. Niu,

We are pleased to inform you that your manuscript has been judged scientifically suitable for publication and will be formally accepted for publication once it complies with all outstanding technical requirements.

With kind regards,

Haitao Shi

Academic Editor

PLOS ONE
---

## [Editor Report · Acceptance letter]

5 May 2020

PONE-D-19-35964R1 

 Gene expression and metabolic changes of Momordica charantia L. seedlings in response to low temperature stress 

Dear Dr. Niu:

I am pleased to inform you that your manuscript has been deemed suitable for publication in PLOS ONE. Congratulations! Your manuscript is now with our production department. 

With kind regards,

on behalf of

Dr. Haitao Shi 

Academic Editor

PLOS ONE